

# A new extension of Poisson distribution for asymmetric count data: theory, classical and Bayesian estimation with application to lifetime data

Abdullah Alomair[1] and Muhammad Ahsan-ul-Haq[2]

[1] Department of Quantitative Methods, School of Business, King Faisal University, Al-Ahsa, Saudi Arabia
[2] College of Statistical Sciences, University of the Punjab, Lahore, Pakistan

## ABSTRACT

Several research investigations have stressed the importance of discrete data analysis and its relevance to actual events. The current work focuses on a new discrete distribution with a single parameter that can be derived using the Poisson mixing technique. The new distribution is named the Poisson Entropy-Based Weighted Exponential Distribution. It is useful for discussing asymmetric "right-skewed" data with "heavy" tails. Its failure rate function can be used to explain situations with increasing failure rates. The statistical properties of the new distribution are expressed explicitly. The proposed model is simple to manage for under-, equal-, and over-dispersed datasets. The model parameters are estimated using the maximum likelihood method. We consider the parameter estimation for the new model based on right-censored data with a cure fraction. One more focus of the present study is the Bayesian estimation of the model parameters. In the end, three real-world dataset examples were utilized to show the value of the new distribution. These applications revealed that the new model outperforms other standard discrete models.

# INTRODUCTION

Numerous studies have emphasized the relevance of count data modeling which has aroused significant interest in a range of fields such as medical research, earth science, physics, economics, and insurance. Various lifetime probability distributions have been utilized and investigated in reliability theory. The Poisson distribution is commonly utilized to analyze the "symmetric" and "asymmetric" count datasets, but it cannot describe over-dispersed datasets. As a result, there has been a lot of interest in the discretization of continuous probability distributions. Several techniques may be used to obtain the discrete analog of a continuous probability distribution. The Poisson mixed approach gets great attention from researchers and is most commonly used for generalization or generation of new probability distributions. The Poisson mixed approach is discussed below.

If the Poisson parameter is a random variable with a parameterized distribution (P), then the resulting model is a discrete Poisson mixed model. The distribution P and its

Corresponding author
Abdullah Alomair,
ama.alomair@kfu.edu.sa

parameter vector $\Theta$ are referred to as prior distribution and hyperparameter, respectively. The resulting distribution of random variable X is stated as follows:

$$f_X(x) = \int_0^\infty f_{X|\Lambda}(X|\lambda) f_\Lambda^P(\lambda) d\lambda, \tag{1}$$

where $X|\Lambda$ is the Poisson distribution with parameter $\lambda$ as

$$f_{X|\Lambda}(X|\lambda) = \frac{e^{-\lambda}\lambda^x}{x!}, \quad x = 0, 1, 2, 3, \ldots . \tag{2}$$

$f_\Lambda^P(\lambda)$ is a continuous density function and $\Lambda$ is a random variable of the Poisson parameter $\lambda$.

In the literature, many authors have compounded the standard Poisson parameter using standard lifetime distributions. The negative binomial distribution was derived by *Greenwood & Yule (1920)* by combining the Poisson and gamma distributions. *Johnson, Kemp & Kotz (1992)* combined the Poisson and exponential distributions to get the geometric distribution. Similarly, various authors introduced mixed Poisson distributions, some examples include the Poisson Lindley (*Sankaran, 1970*), Poisson Pseudo Lindley (*Zeghdoudi & Nedjar, 2017*), Poisson transmuted exponential (*Bhati, Kumawat & Gómez-Déniz, 2017*), Poisson Xgamma (*Altun, Cordeiro & Ristić, 2021*), Poisson Ailamujia (*Hassan et al., 2020*), Poisson Quasi-Lindley (*Altun, 2019*), Poisson XLindley (*Ahsan-ul-Haq et al., 2022*), Poisson moment exponential (*Ahsan-ul-Haq, 2022*), and Poisson Mirra (*Maya et al., 2022*).

*Al-Nasser, Rawashdeh & Talal (2022)* introduced a new weighted exponential distribution. The resulting distribution is named entropy-based weighted exponential distribution (EBWED). Let X be a continuous random variable that follows EBWED with a single parameter $(\beta)$. The probability density function (pdf) of EBWD will be.

$$f(x, \beta) = \frac{\beta(\beta x - \ln(\beta))}{1 - \ln(\beta)} e^{-\beta x}; \qquad x > 0; \ \ln(\beta) \neq 1 \tag{3}$$

The cumulative distribution function (cdf) of the EBWED is

$$F(x, \beta) = 1 - \frac{(1 + \beta x - \ln(\beta))}{1 - \ln(\beta)} e^{-\beta x}. \tag{4}$$

The innovation of this study is the derivation of a new Poisson mixed distribution for under, equal, and over-dispersed count datasets to address the above-mentioned issues. This study has the following goals;

- The main objective is to introduce a new flexible Poisson entropy-based weighted exponential distribution. The ensuing distribution is obtained by mixing Poisson with the entropy-based weighted exponential distribution. The moments and associated measures of the new distribution can be calculated analytically when compared to

existing discrete distributions, and it has a strong modeling capability. The new model is also incredibly adaptable.

- The model parameter is estimated using the maximum likelihood estimation (MLE) method. A comprehensive simulation is performed to assess the behavior ML estimates.
- The new distribution is used to model "asymmetric" and "right skewed" data in the presence of complete and right-censored data.
- We also take into account censored data with a cure fraction.
- The Bayesian estimation approach is also utilized to estimate the model parameter.

The rest of the document is structured as follows: The derivation of the new discrete probability model is presented in "The PEBWE Distribution". "Moments and Associated Measures" discusses its underlying mathematical characteristics. "Parameter Estimation" discusses the maximum likelihood estimation for the distribution parameter using complete, censored, and censored data with cure fraction. This section also discusses Bayesian estimation using the MCMC approach. Three examples are given in "Application" to illustrate the adaptability of the new distribution. In the end, concluding remarks and some future directions are given in "Conclusion".

## THE PEBWE DISTRIBUTION

The following proposition introduces a new mixed-Poisson model by combining the Poisson and Entropy-Based Weighted Exponential distributions.

*Proposition 1.* Suppose that X follows the compound Poisson-EBWE distribution (PEBWED), which has the following stochastic representation:

$(X|\lambda) \sim Poisson\ (\lambda)$
$(\lambda|\beta) \sim EBWE(\beta)$

where $\lambda$ and $\beta > 0$. Then, the pmf of X is given by

$$p(x;\beta) = \Pr(X = x) = \frac{\beta((1+\beta)\ln(\beta) - (1+x)\beta)}{(1+\beta)^{2+x}(\ln(\beta) - 1)}, \qquad x = 0, 1, 2, \tag{5}$$

The new model is denoted as *PEBWED($\beta$)*, and one can note $X \sim PEBWED(\beta)$ to apprise that X follows that PEBWED with parameter $\beta$.

**Proof.** The pmf of X can be obtained using the common mixing method shown below.

$$p(x;\beta) = \int_0^\infty \Pr(X = x|\lambda)f(\lambda|\beta)d\lambda,$$

$$= \int_0^\infty \frac{\lambda^x e^{-\lambda}}{x!} \frac{\beta(\beta\lambda - \ln(\beta))e^{-\beta\lambda}}{1 - \ln(\beta)} d\lambda,$$

$$= \frac{1}{(1 - \ln(\beta))x!} \left\{ \beta^2 \int_0^\infty \lambda^{x+1} e^{-\lambda} e^{-\beta\lambda} d\lambda - \beta\ln(\beta) \int_0^\infty \lambda^x e^{-\lambda} e^{-\beta\lambda} d\lambda \right\},$$

$$= \frac{1}{(1 - \ln(\beta))x!} \left\{ \frac{\beta^2 \Gamma(x+2)}{(\beta+1)^{x+2}} - \frac{\beta \ln(\beta) \Gamma(x+1)}{(\beta+1)^{x+1}} \right\},$$

$$= \frac{\beta((1+\beta)\ln(\beta) - (1+x)\beta)}{(1+\beta)^{2+x}(\ln(\beta) - 1)}, \qquad x = 0, 1, 2, \dots$$

The proof is completed.

Figure 1 depicts the potential pmf plots of the proposed distribution.

**Remark:** The first derivative of pmf is

$$\frac{dp(x)}{dx} = -\frac{\beta(\beta + ((1+\beta)\ln(\beta) - (1+x)\beta)\ln(1+\beta))}{(1+\beta)^{2+x}(\ln(\beta) - 1)},$$

gives

$$\hat{x} = \frac{\beta - \beta \ln(1+\beta) + \ln(\beta)\ln(1+\beta) + \beta \ln(\beta)\ln(1+\beta)}{\beta \ln(1+\beta)}. \tag{6}$$

For $\beta > 0.6934$ the $\hat{X}$ is a critical point that maximizes the $p(\hat{X}; \beta)$ and $0 < \beta \le 0.6914$ the pmf is a decreasing function of x.

and

$$\frac{d^2 p(x)}{dx^2} = \frac{\beta \ln(1+\beta)(2\beta + (-(1+x)\beta + (1+\beta)\ln(\beta))\ln(1+\beta))}{(1+\beta)^{2+x}(\ln(\beta) - 1)}.$$

Therefore, the mode of PEBWED is given by

$$Mode(X) = \begin{cases} \dfrac{\beta - \beta \ln(1+\beta) + \ln(\beta)\ln(1+\beta) + \beta \ln(\beta)\ln(1+\beta)}{\beta \ln(1+\beta)} & \text{for } \beta > 0.6914 \\ 0 & \text{otherwise} \end{cases}$$

The cdf and survival function of the PEBWED is given by

$$F(x; \beta) = \Pr(X \le x) = 1 - \frac{(1 + \beta(2+x) - (1+\beta)\ln(\beta))}{(1+\beta)^{2+x}(1 - \ln(\beta))}, \tag{7}$$

and

$$S(x; \beta) = \frac{(1 + \beta(2+x) - (1+\beta)\ln(\beta))}{(1+\beta)^{2+x}(1 - \ln(\beta))}. \tag{8}$$

The hazard function (hf) of the PEBWED is given by

$$h(x; \beta) = \frac{p(x; \beta)}{1 - F(x; \beta)} = \frac{\beta((-1-x)\beta + (1+\beta)\ln(\beta))}{(1+\beta)\ln(\beta) - (2+x)\beta - 1}. \tag{9}$$

*Proposition 2:* The PEBWED hf increases as x increases.

**Proof:** Using the idea of *Glaser (1980)* and from the pmf of PEBWED

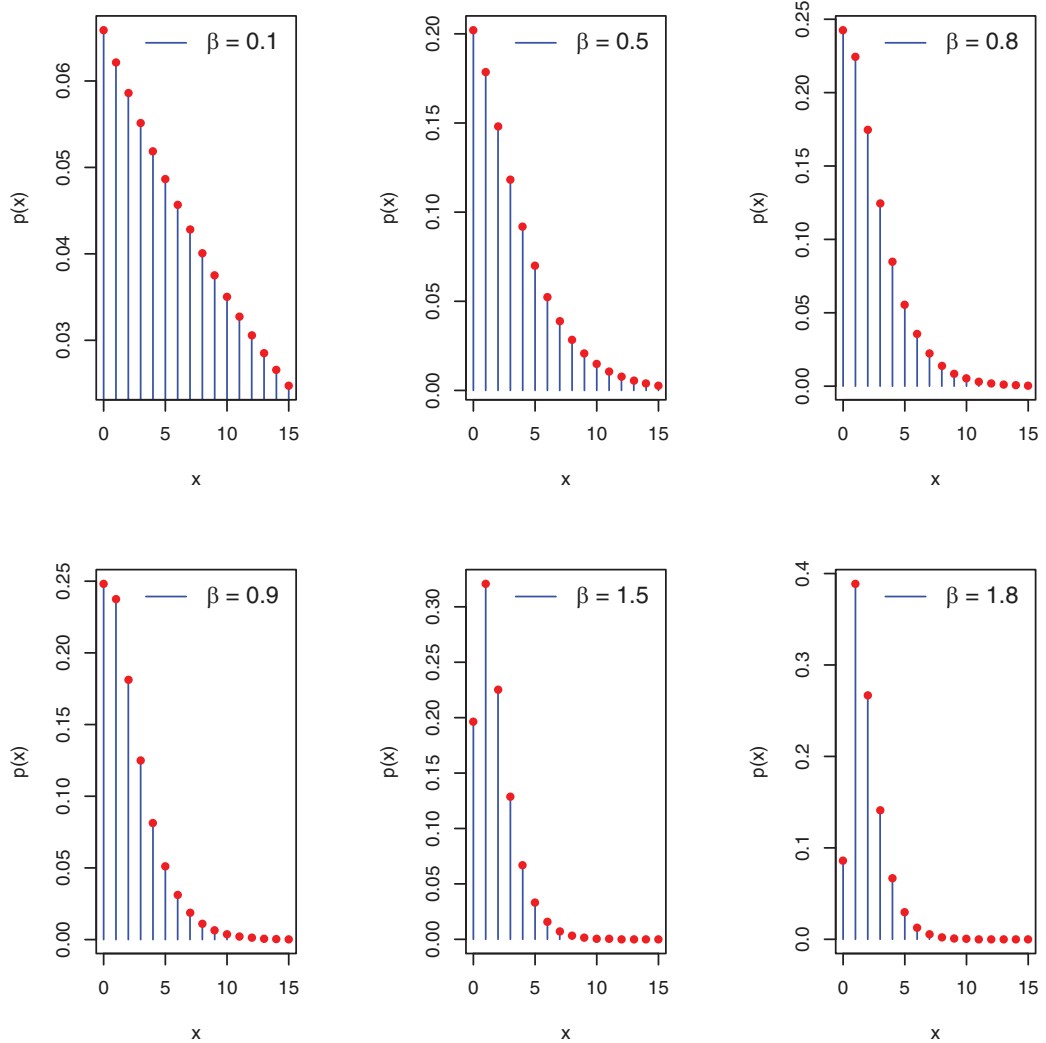

**Figure 1  Plots of pmf of the PEBWED.**     

$$\rho(x) = -\frac{p'(x; \beta)}{p(x; \beta)}$$

$$= -\frac{\beta}{(1+x)\beta - (1+\beta)\ln(\beta)} + \ln(1+\beta)$$

It follows that

$$\rho'(x) = \frac{\beta^2}{((1+x)\beta - (1+\beta)\ln(\beta))^2}.$$

As $\rho'(x) > 0$, the hf of PEBWED is increasing function.
Furthermore, the graphs in Fig. 2 pertain to the possible shapes of the PEBWED.

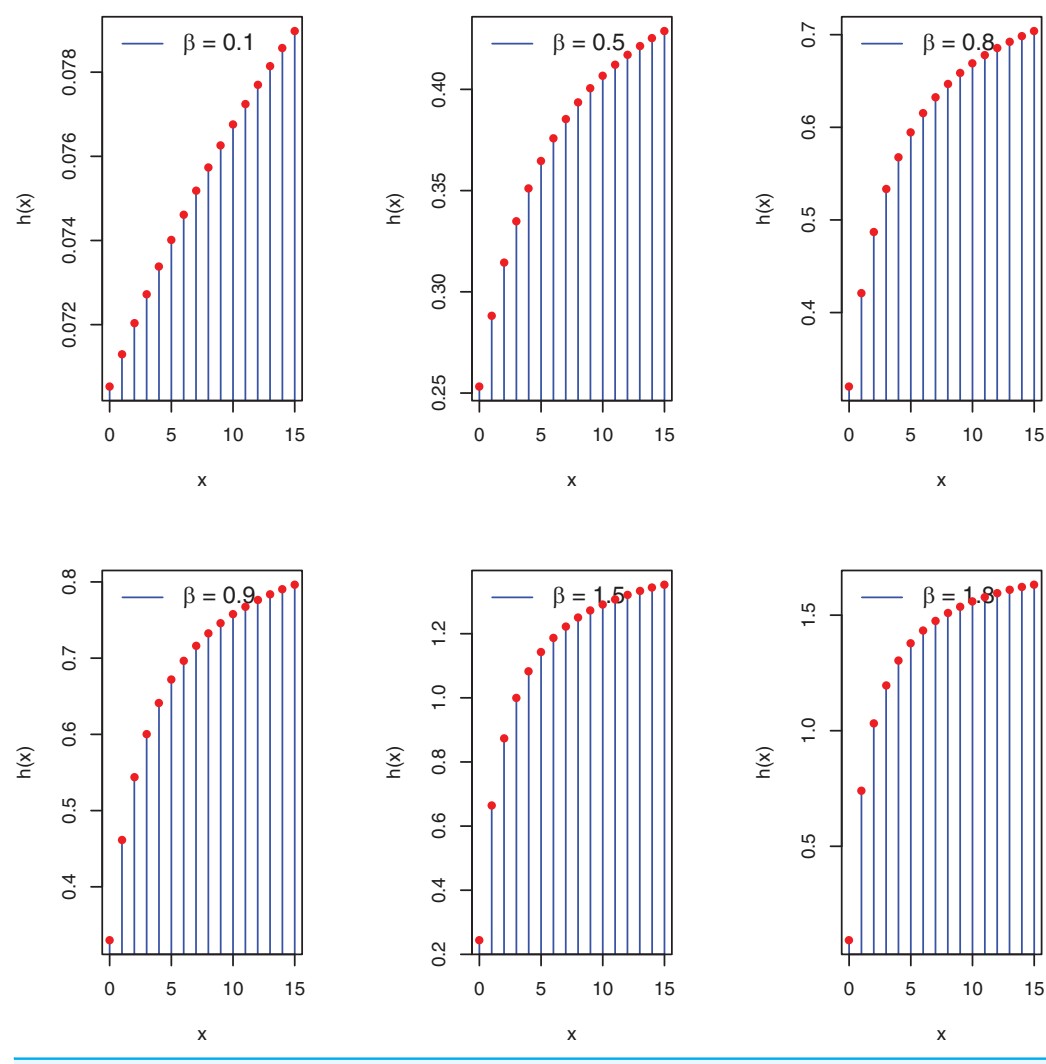

**Figure 2** **Plots of hf of the PEBWED.**

## MOMENTS AND ASSOCIATED MEASURES

In this section, moments, probability generating function, moment generating function, and their associated measures, mean, variance, dispersion index, skewness, and kurtosis are derived and discussed.

*Proposition 3:* The rth factorial moments of PEBWED are given by

$$\mu_{(r)} = E(X(X-1)\ldots(X-r+1)) = \frac{\Gamma(1+r)(\ln(\beta) - r - 1)}{\beta^r(\ln(\beta) - 1)}. \tag{10}$$

**Proof:** The factorial moment can be calculated using the compound-Poisson theory as follows:

$$\mu_{(r)} = \frac{\beta}{1 - \ln(\beta)} \int_0^\infty \lambda^r(\beta\lambda - \ln(\beta))e^{-\beta\lambda}d\lambda,$$

$$= \frac{\beta}{1 - \ln(\beta)} \left\{ \beta \int_0^\infty \lambda^{r+1} e^{-\beta\lambda} d\lambda - \ln(\beta) \int_0^\infty \lambda^r e^{-\beta\lambda} d\lambda \right\},$$

$$= \frac{\Gamma(1 + r)(\ln(\beta) - r - 1)}{\beta^r(\ln(\beta) - 1)}.$$

which complete the proof.

By replacing r = 1, 2, 3, and 4 in Eq. (10), the first four factorial moments of the PEBWED can be derived.

That is,

$$\mu_{(1)} = \frac{\ln(\beta) - 2}{\beta(\ln(\beta) - 1)},$$

$$\mu_{(2)} = \frac{2(\ln(\beta) - 3)}{\beta^2(\ln(\beta) - 1)},$$

$$\mu_{(3)} = \frac{6(\ln(\beta) - 4)}{\beta^3(\ln(\beta) - 1)},$$

and

$$\mu_{(4)} = \frac{24(\ln(\beta) - 5)}{\beta^4(\ln(\beta) - 1)}.$$

Now, using the general connection between factorial moments and moments about the origin, the first four moments about the origin of the PEBWED are obtained. We get

$$\mu_1' = \mathrm{E}(X) = \frac{\ln(\beta) - 2}{\beta(\ln(\beta) - 1)}, \tag{11}$$

$$\mu_2' = \mathrm{E}(X^2) = \frac{-2(3 + \beta) + (2 + \beta)\ln(\beta)}{\beta^2(\ln(\beta) - 1)},$$

$$\mu_3' = \mathrm{E}(X^3) = \frac{-2(12 + \beta(9 + \beta)) + (6 + \beta(6 + \beta))\ln(\beta)}{\beta^3(\ln(\beta) - 1)},$$

$$\mu_4'^i = \mathrm{E}(X^4) = \frac{-2(60 + \beta(72 + \beta(21 + \beta))) + (2 + \beta)(12 + \beta(12 + \beta))\ln(\beta)}{\beta^4(\ln(\beta) - 1)}.$$

Therefore, the variance of PEBWED is obtained as

$$\mathrm{Var(X)} = \frac{2(1 + \beta) + \ln(\beta)(-4 - 3\beta + (1 + \beta)\ln(\beta))}{\beta^2(\ln(\beta) - 1)}. \tag{12}$$

The dispersion Index (DI) of the PEBWED is given by

$$\mathrm{DI(X)} = \frac{2(1 + \beta) + \ln(\beta)(-4 - 3\beta + (1 + \beta)\ln(\beta))}{\beta(\ln(\beta) - 2)(\ln(\beta) - 1)}. \tag{13}$$

To obtain explicit formulations for the skewness and kurtosis of the PEBWED, apply the following equations.

$$Skewness(X) = \frac{E(X^3) - 3E(X^2)E(X) + 2(E(X))^3}{[Var(X)]^{\frac{3}{2}}},$$

and

$$Kurtosis(X) = \frac{E(X^4) - 4E(X^3)E(X) + 6E(X^2)(E(X))^2 - 3(E(X))^4}{[Var(X)]^2}.$$

**Proposition 4:** The probability generating function (pgf) of PEBWED is given by

$$G(s) = E(x^X) = \frac{\beta(-\beta + (1 - s + \beta)\ln(\beta))}{(1 - s + \beta)^2(\ln(\beta) - 1)}. \tag{14}$$

for $s \in (-1, 1)$.

**Proof:** The pgf of the PEBWED is derived using the well-known compound-Poisson theory in the manner described below

$$G(s) = \frac{\beta}{1 - \ln(\beta)} \int_0^\infty e^{\lambda(s-1)}(\beta\lambda - \ln(\beta))e^{-\beta\lambda}d\lambda,$$

$$= \frac{\beta}{1 - \ln(\beta)} \left\{ \beta \int_0^\infty e^{\lambda(s-1)}\lambda e^{-\beta\lambda}d\lambda - \ln(\beta) \int_0^\infty e^{\lambda(s-1)}e^{-\beta\lambda}d\lambda \right\},$$

$$= \frac{\beta}{1 - \ln(\beta)} \left\{ \frac{\beta}{(1 - s + \beta)^2} - \frac{\ln(\beta)}{(1 - s + \beta)} \right\},$$

$$= \frac{\beta(-\beta + (1 - s + \beta)\ln(\beta))}{(1 - s + \beta)^2(\ln(\beta) - 1)}.$$

which completes the proof.

The moment generating function (mgf) and characteristic function (cf) of the PEBWED are obtained from Eq. (14) when s is substituted by $e^t$ and $e^{it}$ respectively. They are provided, respectively, by

$$M(t) = \frac{\beta(\beta - \ln(\beta) + e^t\ln(\beta) - \beta\ln(\beta))}{(-1 + e^t - \beta)^2(1 - \ln(\beta))}. \tag{15}$$

for $t \le 0$, and

$$\phi(t) = \frac{\beta(\beta - \ln(\beta) + e^{it}\ln(\beta) - \beta\ln(\beta))}{(-1 + e^{it} - \beta)^2(1 - \ln(\beta))}. \tag{16}$$

for $t \in \mathbb{R}$.

The mean, variance, DI, skewness, and kurtosis for the PEBWED are now shown numerically in Table 1 for various parameter choices.

**Table 1 Some computational measures of the PEBWED.**

| $\beta$ | E(X) | Var(X) | CS(X) | CK(X) | DI(X) | CV(X) |
|------|--------|--------|--------|--------|--------|--------|
| 0.1 | 13.028 | 164.42 | 1.7974 | 7.6916 | 12.620 | 0.9842 |
| 0.5 | 3.1812 | 10.511 | 1.6455 | 6.8850 | 3.3040 | 1.0191 |
| 0.8 | 2.2720 | 5.3450 | 1.5602 | 6.4932 | 2.3526 | 1.0176 |
| 0.9 | 2.1163 | 4.5742 | 1.5307 | 6.3703 | 2.1614 | 1.0106 |
| 1.2 | 1.8525 | 3.2068 | 1.4354 | 6.0221 | 1.7311 | 0.9667 |
| 1.5 | 1.7880 | 2.4702 | 1.3414 | 5.7724 | 1.3815 | 0.8790 |
| 1.8 | 1.9033 | 1.8930 | 1.3640 | 5.9838 | 0.9946 | 0.7229 |
| 2.0 | 2.1294 | 1.3538 | 2.0214 | 7.3516 | 0.6358 | 0.5464 |

# PARAMETER ESTIMATION

In this section, the model parameter is estimated using the maximum likelihood approach based on complete and censored sampling, censored sampling with cure fraction. This section also covers parameter estimation using the Bayesian approach.

## ML estimation based on complete data

Let $X_1, X_2, \ldots, X_n$ be a random sample obtained from a PEBWE distribution. The log-likelihood function is defined as follows

$$l(\beta|x) = n\log(\beta) - n\log(1 - \ln(\beta)) + \sum_{i=1}^{n}\log(\beta x_i - \ln(\beta)) + \beta\sum_{i=1}^{n}x_i, \tag{17}$$

For the maximum likelihood (ML) estimator of the parameter, differentiate Eq. (17) for $\beta$

$$\frac{\partial l(\beta|x)}{\partial\beta} = \frac{n}{\beta} + \frac{n}{1 - \log(\beta)} + \sum_{i=1}^{n}\frac{x_i - \frac{1}{\beta}}{\beta x_i - \ln(\beta)} + \sum_{i=1}^{n}x_i. \tag{18}$$

Equating Eq. (18) to zero and solving for yields the ML estimator. The resultant expression has no closed-form solution, implying that numerical methods are required to get the ML estimate of the parameter.

## ML estimation based on censored data

Given a random sample $(x_i, \ d_i)$ of size $n, \ i = 1, \ldots, n$, the ith individual's involvement to the likelihood function is given by

$$L_i = [f(x_i)]^{d_i}[S(x_i)]^{1-d_i},$$

where $d_i$ is a censoring indicator variable; it is equal to one for the survival time that was observed and zero for one that was right-censored. The likelihood function for the model parameter is provided by when the data have a PEBWE distribution.

$$L(\beta|x,d) = \prod_{i=1}^{n} \left[ \frac{\beta(\beta x_i - \ln(\beta))}{1 - \ln(\beta)} e^{-\beta x_i} \right]^{d_i} \left[ \frac{(1 + \beta x_i - \ln(\beta))}{1 - \ln(\beta)} e^{-\beta x_i} \right]^{1-d_i}, \qquad (19)$$

The corresponding loglikelihood function is

$$l(\beta|x,d) = \log(\beta) \sum_{i=1}^{n} d_i - \log(1 - \ln(\beta)) \sum_{i=1}^{n} d_i + \sum_{i=1}^{n} d_i \log(\beta x_i - \ln(\beta)) - \sum_{i=1}^{n} d_i \beta x_i$$
$$+ \sum_{i=1}^{n} (1 - d_i) \log(1 + \beta x_i - \ln(\beta)) - \sum_{i=1}^{n} (1 - d_i) \log(1 - \ln(\beta)) - \sum_{i=1}^{n} (1 - d_i) \beta x_i. \qquad (20)$$

We have derived the log-likelihood function about $\beta$

$$\frac{\partial l(\beta|x,d)}{\partial \beta} = \sum_{i=1}^{n} \frac{d_i}{\beta} + \sum_{i=1}^{n} \frac{\frac{d_i}{\beta}}{1 - \ln(\beta)} + \sum_{i=1}^{n} \frac{d_i \left( x_i - \frac{1}{\beta} \right)}{\beta x_i - \ln(\beta)} - \sum_{i=1}^{n} d_i x_i$$
$$+ \sum_{i=1}^{n} \frac{(1 - d_i) \left( x_i - \frac{1}{\beta} \right)}{1 + \beta x_i - \ln(\beta)} + \sum_{i=1}^{n} \frac{(1 - d_i) \left( \frac{1}{\beta} \right)}{1 - \ln(\beta)} - \sum_{i=1}^{n} (1 - d_i) x_i. \qquad (21)$$

When we set Eq. (21) to zero, we have the scoring equation that corresponds, and its numerical solution yields the ML estimator.

## ML estimation based on censored data and a cure fraction

Survival analysis reveals that a subset of people seems to be impervious to the occurrence of the important event. In clinical trials, some patients who react to the treatment may experience prolonged symptom relief or perhaps a complete recovery. The conventional mixing model's survival function is provided by

$$S(x) = \eta + (1 - \eta) S_0(x),$$

where $\eta \in (0, 1)$ is the proportion of immunes or cure fraction, and $S_0(x)$ is a baseline survival function for vulnerable persons. Given a random sample $(x_i, d_i)$ of size $n$, $i = 1; \ldots, n$, the $i^{\text{th}}$ subject's contribution to the likelihood function is given by

$$L_i = [f(x_i)]^{d_i} [S(x_i)]^{1-d_i} = [(1 - \eta) f_0(x)]^{d_i} [\eta + (1 - \eta) S_0(x)]^{1-d_i},$$

where $f_0(x)$ is the susceptible individuals' baseline pdf and $d_i$ is a censoring indicator variable. The likelihood and log-likelihood functions for parameter $\beta$ are given below.

$$L(\beta, \eta|x,d) = \prod_{i=1}^{n} \left[ (1-\eta) \frac{\beta(\beta x_i - \ln(\beta))}{1 - \ln(\beta)} e^{-\beta x_i} \right]^{d_i} \left[ \eta + (1-\eta) \frac{(1 + \beta x_i - \ln(\beta))}{1 - \ln(\beta)} e^{-\beta x_i} \right]^{1-d_i}, (22)$$

and

$$l(\beta, \eta | x, d) = \sum_{i=1}^{n} d_i \log\left((1 - \eta)\frac{\beta(\beta x_i - \ln(\beta))}{1 - \ln(\beta)} e^{-\beta x_i}\right)$$
$$+ \sum_{i=1}^{n}(1 - d_i)\log\left(\eta + (1 - \eta)\frac{(1 + \beta x_i - \ln(\beta))}{1 - \ln(\beta)} e^{-\beta x_i}\right). \tag{23}$$

After differentiating the log-likelihood function for parameters and setting the resultant derivatives to zero, the ML estimators are generated by solving the appropriate equations.

## Bayesian estimation

The Bayesian approach has become the most extensively utilized technique in a range of domains, including but not limited to numerous applications. It is especially helpful in engineering, reliability, health sciences, epidemiology, and quality studies due to its capacity to incorporate prior information into the study. So, under this approach, a prior distribution must be assigned to each parameter. For the PEBWE distribution, we can consider the gamma distribution as the prior distribution for the parameter $\beta$ and the beta distribution for the cure fraction parameter $\eta$. The density functions for the gamma and beta distributions are

$$\beta \sim \Gamma(\tau_1, \lambda_1), \quad \tau_1, \lambda_1 > 0,$$

and

$$\eta \sim \Gamma(\tau_2, \lambda_2), \quad \tau_2, \lambda_2 > 0.$$

where $\tau_1, \lambda_1, \tau_2, \lambda_2$ are the hyperparameters.

The joint posterior expression is gained by multiplying the likelihood function given in Eq. (17) by the prior distribution densities. To simulate the sample from the posterior density, we utilized the Markov chain Monte Carlo (MCMC) procedures as Gibs sampling. We generate 1,006,000 samples for each denomination of parameter. The first 6,000 simulated samples were eliminated as part of a burn-in phase, which is often used to reduce the influence of starting values. The parameter Bayesian estimates were obtained as the mean of samples specified from the joint posterior distribution. Traceplots and the Geweke diagnostic were used to monitor the convergence of the simulated samples. Further, the highest posterior density (HPD) interval of 95% was obtained using the simulated posterior distributions.

## Simulation

Here, we conduct a comprehensive simulation analysis to assess the maximum likelihood estimation approach using complete data. Random samples of the PEBWE distribution of sizes (n) 10, 20, 50, 100, and 200 were used considering different values of the parameter ($\beta$). All simulation results were based on N = 10,000 replications for the different sample sizes considered for each parameter setting. Table 2 shows the results of the average estimates, absolute bias (AB), mean relative error (MRE), and mean square error (MSE) of all parameter values.

| Table 2 Simulation results based on complete data. | | | | |
|---|---|---|---|---|
| Parameter | n | AB | MRE | MSE |
| $\beta = 0.1$ | 10 | 0.0141 | 0.1410 | 0.0024 |
| | 20 | 0.0064 | 0.0642 | 0.0008 |
| | 50 | 0.0024 | 0.0245 | 0.0003 |
| | 100 | 0.0010 | 0.0102 | 0.0001 |
| | 200 | 0.0007 | 0.0074 | 0.0001 |
| $\beta = 0.5$ | 10 | 0.1533 | 0.3067 | 0.1955 |
| | 20 | 0.0660 | 0.1320 | 0.0531 |
| | 50 | 0.0180 | 0.0360 | 0.0106 |
| | 100 | 0.0106 | 0.0212 | 0.0046 |
| | 200 | 0.0049 | 0.0097 | 0.0021 |
| $\beta = 0.8$ | 10 | 0.1794 | 0.2242 | 0.2393 |
| | 20 | 0.1020 | 0.1275 | 0.1153 |
| | 50 | 0.0480 | 0.0600 | 0.0420 |
| | 100 | 0.0219 | 0.0273 | 0.0175 |
| | 200 | 0.0117 | 0.0147 | 0.0078 |
| $\beta = 0.9$ | 10 | 0.1597 | 0.1774 | 0.2351 |
| | 20 | 0.1032 | 0.1146 | 0.1186 |
| | 50 | 0.0522 | 0.0580 | 0.0502 |
| | 100 | 0.0239 | 0.0266 | 0.0237 |
| | 200 | 0.0112 | 0.0124 | 0.0107 |
| $\beta = 1.5$ | 10 | 0.0309 | 0.0206 | 0.2654 |
| | 20 | 0.0326 | 0.0218 | 0.1008 |
| | 50 | 0.0280 | 0.0187 | 0.0433 |
| | 100 | 0.0152 | 0.0102 | 0.0223 |
| | 200 | 0.0108 | 0.0072 | 0.0110 |
| $\beta = 1.8$ | 10 | 0.2553 | 0.1418 | 0.4160 |
| | 20 | 0.1028 | 0.0571 | 0.1776 |
| | 50 | 0.0012 | 0.0007 | 0.0192 |
| | 100 | 0.0035 | 0.0020 | 0.0030 |
| | 200 | 0.0012 | 0.0007 | 0.0014 |

## APPLICATION

In this section, the new model is applied to three over-dispersed and asymmetric, and right-skewed datasets. We compare the fits of PEBWE distribution with Poisson Ailamujia (PA), discrete Burr Hatke (DBH), discrete inverted Topp-Leone (DITL), discrete moment exponential (DME), and Poisson distributions. Different model selection and goodness-of-fit criteria, log-likelihood (L), Akaike information criteria (AIC), Bayesian information criteria (BIC), and Kolmogorov-Smirnov tests are used to compare the fitted models.

**Data I:** The first data set is about the number of daily death due to coronavirus in China from 23 January to 28 March 2020. The data set is reported at https://www.worldometers. info/coronavirus/country/china/. The data are: 8, 16, 15, 24, 26, 26, 38, 43, 46, 45, 57, 64,

**Table 3 The MLEs and model selection measures for the first dataset.**

| Statistic | Model | | | | | |
|---|---|---|---|---|---|---|
| | PEBWE | PA | DBH | DITL | DME | Poisson |
| $\hat{\beta}$ | 0.02446 | 0.02010 | 0.99974 | 0.35393 | 25.121 | 49.742 |
| SE | 0.00305 | 0.00178 | 0.00185 | 0.04357 | 2.1865 | 0.86814 |
| $-l$ | 324.30 | 329.99 | 461.02 | 366.91 | 330.52 | 1,409.8 |
| AIC | 650.60 | 661.97 | 924.04 | 735.81 | 663.03 | 2,821.6 |
| BIC | 652.79 | 664.16 | 926.23 | 738.00 | 665.22 | 2,823.8 |
| KS | 0.0876 | 0.1670 | 0.8120 | 0.3290 | 0.1720 | 0.4970 |
| $p$-value | 0.6900 | 0.0490 | 0.0000 | 0.0000 | 0.0410 | 0.0000 |

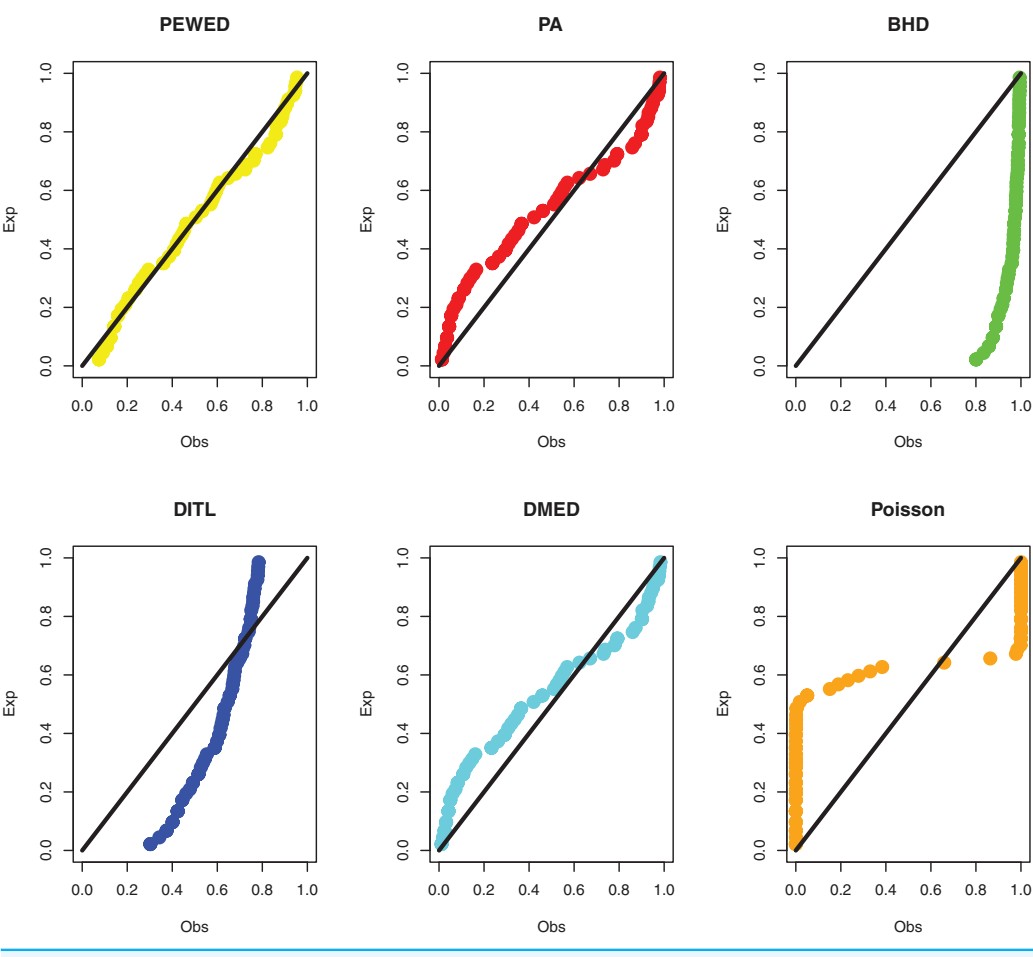

**Figure 3 PPP plots of all fitted models for first dataset.**

65, 73, 73, 86, 89, 97, 108, 97, 146, 121, 143, 142, 105, 98, 136, 114, 118, 109, 97, 150, 71, 52, 29, 44, 47, 35, 42, 31, 38, 31, 30, 28, 27, 22, 17, 22, 11, 7, 13, 10, 14, 13, 11, 8, 3, 7, 6, 9, 7, 4, 6, 5, 3 and 5. The MLEs, standard errors, and goodness-of-fit measures are presented in Table 3. PP plots of all considered distributions for the first dataset are given in Fig. 3.

**Peer**J Computer Science

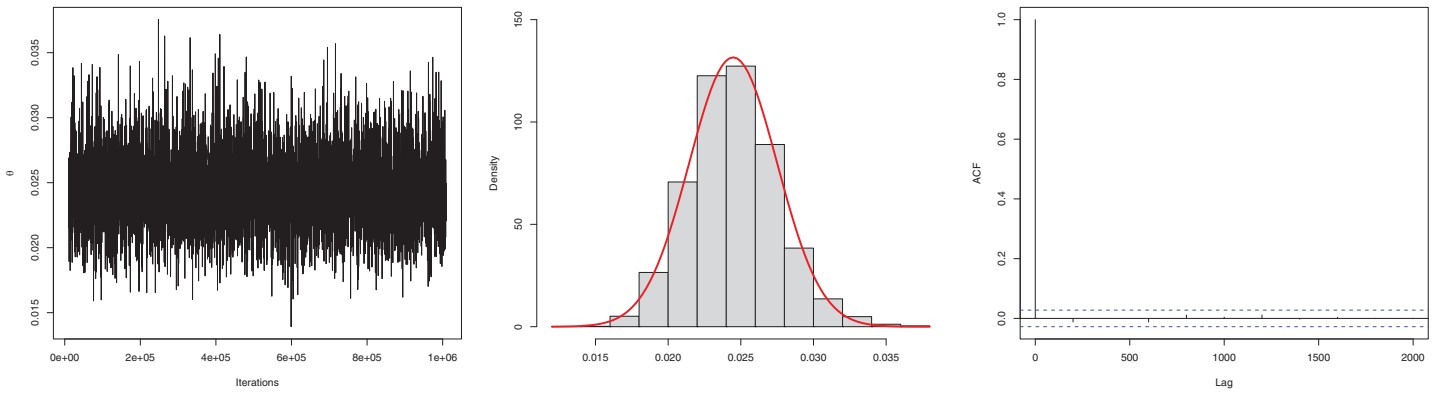

**Figure 4 Traceplot, density, and ACF plot for the first data.**

**Table 4 The MLEs and model selection measures for the second dataset.**

| Statistic | Model | | | | | |
|---|---|---|---|---|---|---|
| | **PEBWE** | **PA** | **DBH** | **DITL** | **DME** | **Poisson** |
| $\hat{\beta}$ | 0.0402 | 0.0352 | 0.9992 | 0.3829 | 14.420 | 19.118 |
| SE | 0.0083 | 0.0050 | 0.0035 | 0.0766 | 1.6500 | 0.8940 |
| $-l$ | 111.21 | 117.44 | 147.97 | 115.24 | 118.19 | 290.41 |
| AIC | 224.41 | 236.88 | 297.94 | 232.48 | 238.38 | 582.81 |
| BIC | 225.81 | 238.29 | 299.34 | 233.88 | 239.78 | 584.21 |
| KS | 0.1630 | 0.2490 | 0.7280 | 0.2650 | 0.2550 | 0.4160 |
| $p$-value | 0.4000 | 0.0480 | 0.0000 | 0.0300 | 0.0400 | 0.0000 |

The next goal of this study was to estimate the model parameter using the Bayesian estimation approach presented in "Bayesian Estimation". The posterior mean for the parameter $\beta$ is 0.0245, and the 95% HPD is 0.0186 to 0.0304. The posterior samples are presented in Fig. 4. The ACF (autocorrelation function) indicates that the posterior samples are independent, and the traceplot demonstrates the appraisal of MCMC samples over the iterations. The Geweke z-score (0.6071) is also indicative of satisfactory convergence of drawn samples to a stable distribution.

**Data II:** The second dataset below is remission times (in weeks) for a group of 30 patients with leukemia who received similar treatment (*Lawless, 2011*). The data observations are; 1, 1, 2, 4, 4, 6, 6, 6, 7, 8, 9, 9, 10, 12, 13, 14, 18, 19, 24, 26, 29, 31+, 42, 45+, 50+, 57, 60, 71+, 85+, 91. The observations with "+" indicate censored times. Using the methodology outlined in "ML estimation based on censored data", we compute the MLEs. Table 4 shows the ML estimate and goodness of fit metrics. Figure 5 shows a comparison of the PP plots for the model based on the PEBWE distribution and the competitive discrete

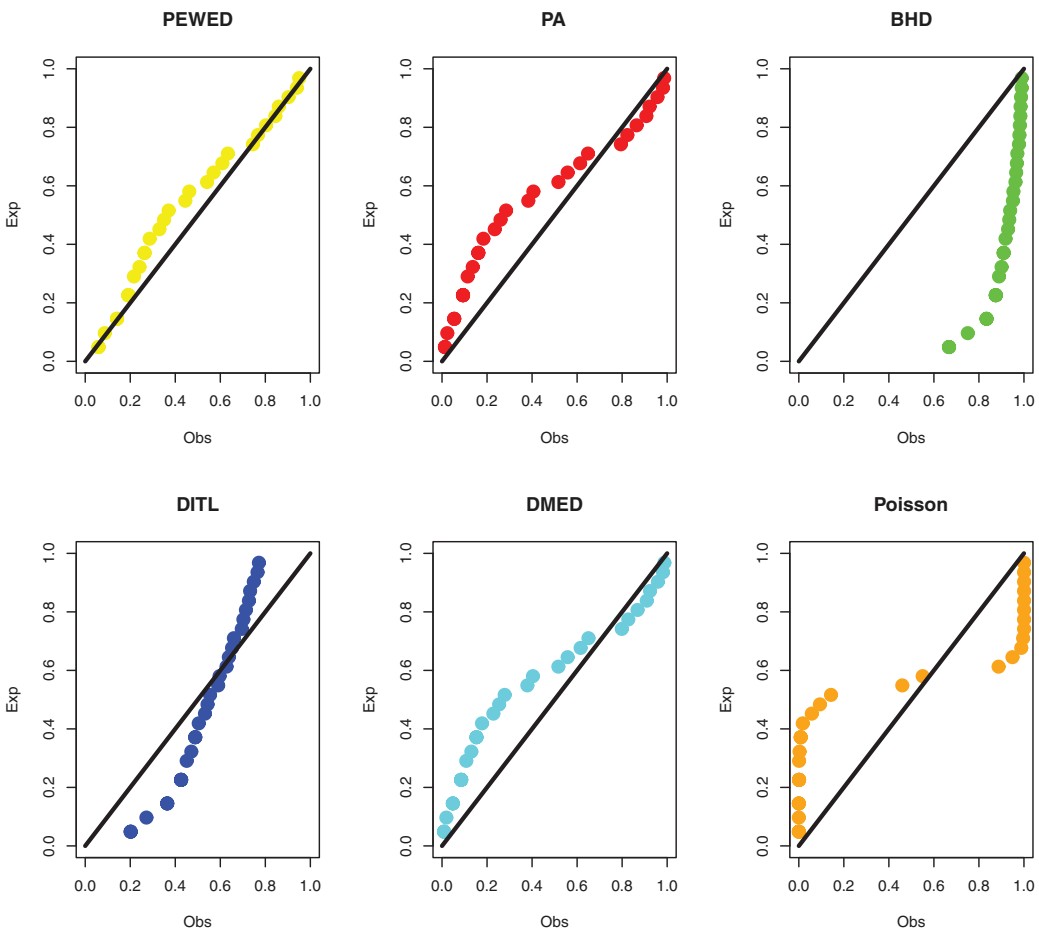

**Figure 5 PP plots of all fitted models for the second dataset.**

distributions. The findings presented by models show that the PEBWE distribution efficiently evaluated this data, while the PA distribution is the second-best model. The data was not well fit by models based on the discrete DBH, DITL, and Poisson distributions.

In the Bayesian estimation, similar to the previous example, we utilized gamma as the prior distribution. The mean is 0.0402, and the 95% HPD is 0.02502 to 0.0565. The posterior samples for the parameter are presented in Fig. 6. The ACF (autocorrelation function) indicates that the posterior samples are independent, and the traceplot demonstrates the appraisal of MCMC samples over the iterations. The Geweke z-score (−0.2249) is also indicative of satisfactory convergence of drawn samples to a stable distribution.

**Data III:** The third dataset is about survival data with a cure fraction. Consider the findings of research done between 2003 and 2013 at the Musculoskeletal Oncology Center of Sun Yat-Sen University's First Affiliated Hospital in China (*Wang et al., 2015*). This study's goal was to assess the efficacy of modular hemipelvis endoprosthesis rebuilding after pelvic tumor resection. Recurrence times for pelvic tumors with marginal or intracapsular margins were 3, 7, 11*, 18, 22*, 25, 28, 32*, 34*, 35, 35*, 36*, 40*, 40*, 41, 54*,

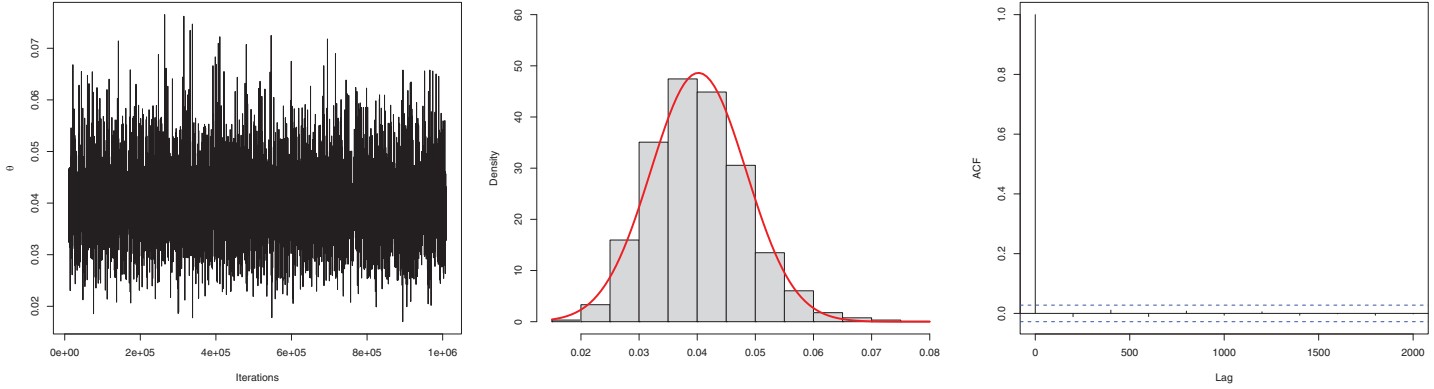

**Figure 6  Traceplot, density, and ACF plot for the second data set.**

**Table 5  The MLEs and model selection measures for the third dataset.**

| Statistic | Model | | | | | |
|---|---|---|---|---|---|---|
| | **PEBWE** | **PA** | **DBH** | **DITL** | **DME** | **Poisson** |
| $\hat{\beta}$ | 0.0307 | 0.0350 | 0.9999 | 0.1130 | 14.369 | 21.922 |
| SE | 0.0264 | 0.0131 | 0.0144 | 0.0883 | 3.9690 | 1.7400 |
| $\hat{\eta}$ | 0.5162 | 0.5799 | 0.6580 | 5.5e-07 | 0.5820 | 0.1290 |
| SE | 0.2279 | 0.1396 | 0.1056 | 0.6640 | 0.1340 | 0.1270 |
| $-l$ | 40.503 | 41.079 | 54.416 | 42.969 | 41.091 | 50.960 |
| AIC | 85.006 | 86.158 | 112.83 | 89.938 | 85.182 | 105.92 |
| BIC | 87.095 | 88.247 | 114.92 | 92.028 | 88.271 | 108.01 |
| KS | 0.2350 | 0.3290 | 0.8410 | 0.6470 | 0.3350 | 0.6500 |
| $p$-value | 0.2000 | 0.0210 | 0.0000 | 0.0000 | 0.0180 | 0.0000 |

$66^*$, $76^*$, $84^*$, $88^*$, and $92^*$ months, with an asterisk (*) denoting a censored observation. We acquire the ML estimations using the approach described in "ML estimation based on censored data and a cure fraction". Table 5 shows the ML estimate and goodness of fit metrics. The PP plots based on all competitive distributions are given in Fig. 7. We can see that the results from the PEBWE distribution provide the best fit.

Similar to the previous example, for the Bayesian estimation, we utilized gamma and beta distribution as prior for $\beta$ and $\eta$ parameters. The means of posterior density for the parameters are $\hat{\beta} = 0.083$ with a 95% HPD interval (0.0301–0.1401) and $\hat{\alpha} = 0.6363$ with a 95% interval (0.4131–0.8434). The posterior samples for the parameter are presented in Fig. 8. The ACF (autocorrelation function) indicates that the posterior samples are independent, and the traceplot demonstrates the appraisal of MCMC samples over the iterations. The Geweke z-score (0.6607) is also indicative of satisfactory convergence of drawn samples to a stable distribution.

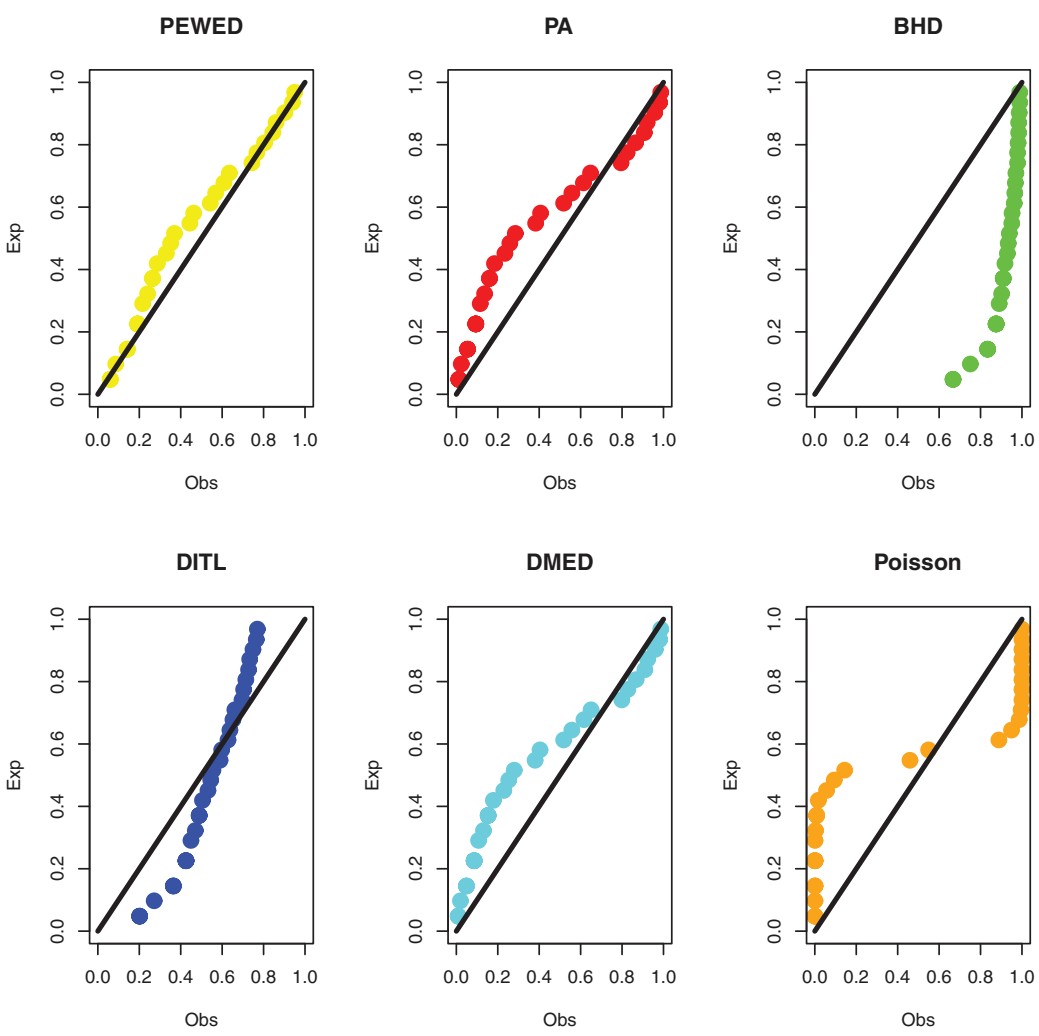

**Figure 7 PP plots of all fitted models for third dataset.**

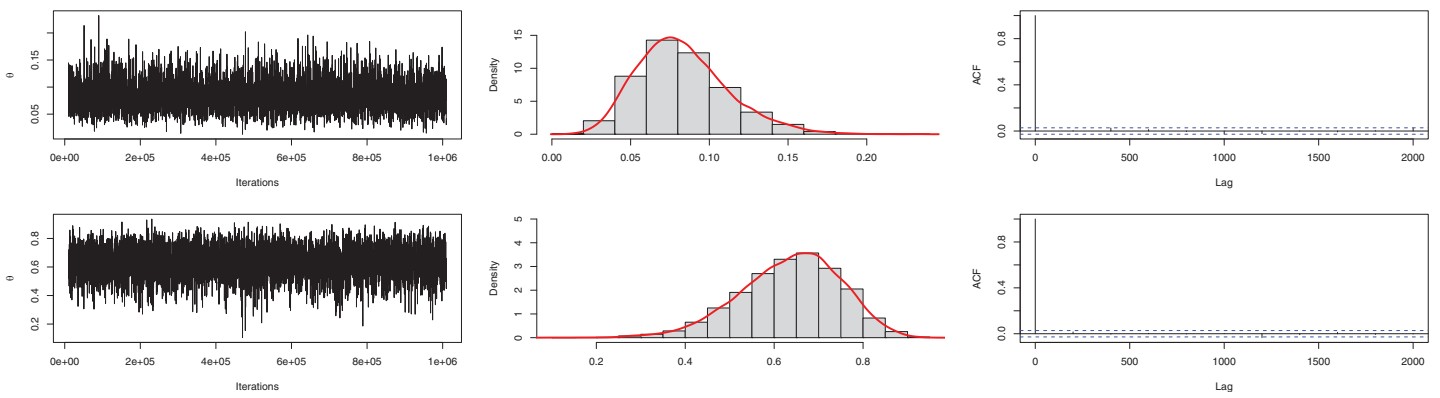

**Figure 8 Traceplot, density, and ACF plot for the third data.**

## CONCLUSION

Discrete probability models play an important role in the analysis of count datasets. A new one-parameter discrete distribution is proposed by mixing Poisson and entropy-based weighted exponential distributions. Derived some important mathematical properties of the new model. The model parameter is estimated using the maximum likelihood and Bayesian estimation methods. The Bayesian estimation was performed using the MCMC approach using the Metropolis-Hastings algorithm. More importantly, the new probability model is applied to three datasets one is based on the number of deaths due to COVID-19, second leukemia patients, and third pelvic tumors patients. The proposed distribution provides more efficient results than all considered competitive distributions.

### Funding

This work was supported by the Deanship of Scientific Research, Vice Presidency for Graduate Studies and Scientific Research, King Faisal University, Saudi Arabia [Grant No. 3479]. The funders had no role in study design, data collection and analysis, decision to publish, or preparation of the manuscript.

### Grant Disclosures

The following grant information was disclosed by the authors:
King Faisal University, Saudi Arabia: 3479.

### Competing Interests

The authors have no conflict of interest.

### Author Contributions

- Abdullah Alomair analyzed the data, authored or reviewed drafts of the article, and approved the final draft.
- Muhammad Ahsan-ul-Haq conceived and designed the experiments, performed the experiments, performed the computation work, prepared figures and/or tables, authored or reviewed drafts of the article, and approved the final draft.

### Data Availability

The R-codes that were used for the computation are available in the Supplemental Files.

The first dataset is available at Worldometer: https://www.worldometers.info/coronavirus/country/china/.

The remission times (in weeks) for a group of 30 patients with leukemia who received similar treatment are: 1, 1, 2, 4, 4, 6, 6, 6, 7, 8, 9, 9, 10, 12, 13, 14, 18, 19, 24, 26, 29, 31+, 42, 45+, 50+, 57, 60, 71+, 85+, 91.

The last dataset is the survival data with a cure fraction. Recurrence times for pelvic tumors with marginal or intracapsular margins were 3, 7, 11*, 18, 22*, 25, 28, 32*, 34*, 35,

35*, 36*, 40*, 40*, 41, 54*, 66*, 76*, 84*, 88*, and 92* months, with an asterisk (*) denoting a censored observation.

## Supplemental Information

Supplemental information for this article can be found online at http://dx.doi.org/10.7717/peerj-cs.1748#supplemental-information.

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
