# Peer review of "A new extension of Poisson distribution for asymmetric count data: theory, classical and Bayesian estimation with application to lifetime data"

_PeerJ Computer Science, doi:10.7717/peerj-cs.1748_

## Round 0.1 · original submission · Major Revisions

I agree with Reviewer #1 and the authors must address the questions related to the fundamental condition that a probability mass function (pmf) must satisfy. The remaining findings must be adjusted based on the modified pmf.

**Language Note:** The review process has identified that the English language must be improved. PeerJ can provide language editing services - please contact us at copyediting@peerj.com for pricing (be sure to provide your manuscript number and title). Alternatively, you should make your own arrangements to improve the language quality and provide details in your response letter. – PeerJ Staff

Reviewer 1 ·

Basic reporting

not satisfied

Experimental design

not a valid methodology and design

Validity of the findings

Fundamental theoretical flaws which nullify all findings and propositions.

Annotated reviews are not available for download in order to protect the identity of reviewers who chose to remain anonymous.

·

Basic reporting

The manuscript is well written and well organized. The derivations of the estimates are correct.

Experimental design

The proposed model was compared with Poisson Ailamujia (PA), discrete Burr Hatke
(DBH), discrete inverted Topp-Leone (DITL), discrete moment exponential (DME), and Poisson
distributions.

Validity of the findings

Different model selection and goodness-of-fit criteria, Log-likelihood (L), Akaike information criteria (AIC), Bayesian information criteria (BIC), and Kolmogorov-Smirnov tests are used to compare the fitted models.

Additional comments

Overall, this is an excellent article.

---

## Round 0.2 · Minor Revisions

Dear Author(s),

Our reviewers and I agree that your manuscript is publishable with minor corrections.

I appreciate your patience in this regard, and I'll be looking forward to the next version.

Thanks for your patience.

Kumer

Reviewer 1 ·

Basic reporting

Well written with necessary literature review/references. Sufficient mathematical proof, analysis, real life data sets are presented. Overall, it's going to be a good new article. I appreciate, authors have been graciously provided their responses to my first review.

Experimental design

Rigorous investigation performed with comparative analysis of the mode with PA, DBH, Poisson etc. Model selection and goodness of fit criteria satisfactory. Sufficient graphical representation and analysis given with explanation.

Validity of the findings

Conclusion, interpretations, theoretical proofs, and tabulated results have been checked and satisfactory.

Additional comments

Line 144 and 145: Please correct the alignment and it should be in one line.
Line 90: It's better to write the pmf in same mathematical form as like line 83.
Figure 3: In all PPP plots, can you use different color and show data values as dots?

Overall, I appreciate the hard work you put into this novel article. Thanks again to both of the authors for the prompt responses to the first revision.

Once you address these few things above, it can be recommended to publish.

·

Basic reporting

pass

Experimental design

pass

Validity of the findings

pass

---

## Round 0.3 · accepted · Accept

Congratulations!

This is an important scholarly work, and we appreciate your patience throughout the review process. Please let us know if you have any questions for us.